# Growth Kinetics and Some Mechanical Properties of Plasma Paste Borided Layers Produced on Nimonic 80A-Alloy

**DOI:** 10.3390/ma14185146

**Published:** 2021-09-08

**Authors:** Natalia Makuch, Piotr Dziarski, Michał Kulka, Mourad Keddam

**Affiliations:** 1Institute of Materials Science and Engineering, Poznan University of Technology, Pl. M.Sklodowskiej-Curie 5, 60-965 Poznan, Poland; natalia.makuch@put.poznan.pl (N.M.); piotr.dziarski@put.poznan.pl (P.D.); 2Laboratoire de Technologie des Matériaux, Faculté de Génie Mécanique et Génie des Procédés, Université des Sciences et de la Technologie Houari Boumediene, BP 32 Bab Ezzouar, Algiers 16111, Algeria; keddam@yahoo.fr

**Keywords:** plasma paste boriding, hardness, wear resistance, cohesion

## Abstract

Plasma paste boriding was employed in order to produce the boride layers on Nimonic 80A-alloy. The process was carried out at temperatures of 1023 K, 1073 K and 1123 K for 3, 4 and 6 h in a gas mixture of 50% H_2_-50% Ar. Borax paste was used as a boron source. The microstructure of the produced surface layers consisted of the mixture of nickel borides and chromium borides. The effect of processing temperature and duration on the thickness of the borided layers was observed. The theoretical thicknesses of the borided layers were estimated using an integral diffusion model. A good correlation was obtained between the theoretical (modeled) and experimental depths of the plasma paste borided layers. The boride layers were characterized by a high hardness ranging from 1160 HV to 2132 HV. The multiphase character of the produced layers resulted in differences in hardness. A significant improvement of the wear resistance of the plasma paste borided Nimonic 80A-alloy was observed in comparison with the non-borided alloy.

## 1. Introduction

Nickel alloys, including Nimonic 80A, are commonly known for their excellent resistance to oxidation and corrosion. However, the poor wear resistance, as an important disadvantage, causes the limited use of these alloys. Under conditions of appreciable mechanical wear (abrasive or adhesive), such materials have to be characterized by suitable wear protection. One of the techniques, which can be used in order to obtain their improved wear resistance, is the surface treatment.

Plasma-assisted surface treatment is a thermo-chemical treatment technique with great potential for industrial application. Due to the high energy of the plasma source, a relatively high efficiency of plasma-assisted processes is ensured. When compared to the conventional thermo-chemical boriding processes, plasma-assisted boriding has many advantages: shorter duration of process, lower temperature of process, relatively low gas consumption, lower activation energy for the formation of the boride layers, and reduction in emission of pollutants and toxic wastes [1,2]. Plasma-assisted boriding has become an interesting technique for producing the borided layers on various materials such as: steels [1,2,3,4,5,6,7], titanium alloys [8,9,10,11,12,13], nickel alloys [14,15], molybdenum alloys [16] and cobalt alloys [17].

An interesting variant of plasma-assisted boriding was the plasma paste boriding process, in which the boron source was a component of paste, and the gases, used in process (e.g., argon, nitrogen), have an inert characteristic [1,9,12,13,14,15]. Titanium and its alloys could be successfully borided using the plasma paste boriding technique [9,12,13]. It was found that the plasma paste boriding processes required a lower activation energy for the formation of the boride layer on the pure titanium or titanium alloys in comparison to the conventional diffusion boriding processes [9]. The gas mixture consisted of 70% hydrogen and 30% argon, and the process was carried out in the temperature range of 973–1073 K for 3, 5 and 7 h [9]. The use of borax paste as a boron source resulted in the formation of dual-phase boride layer (TiB_2_ + TiB) of a high hardness of 2077–2373 HK. The paste used for plasma paste boriding could also contain amorphous boron, as was reported in the case of boriding of AISI 304 stainless steel [1]. It was proved that the activation energy for the formation of boride layer on this steel was 123 kJ·mol^−1^, and was lower in comparison with the conventional diffusion boriding. Multiphase layers contained iron borides, nickel borides and chromium borides and were characterized by a high hardness ranging from 1800 HV to 2000 HV [1]. The activation energy, calculated for the plasma paste boriding of steel and titanium, indicated that this treatment technique ensured the possibility of thick boride layers forming at lower temperatures and shorter duration. Nickel and its alloys are materials which could be also easily subjected to diffusion boriding using different methods. Therefore, in previous works [14,15] the plasma paste boriding was also applied to produce the boride layers on such materials. The substrate materials were as follows: Nickel 201, Inconel 600-alloy and Nimonic 80A-alloy. The chemical composition of Ni-based alloy influenced the phase composition and properties of the produced layers. The increase in Cr content in substrate material diminished the average thickness of the boride layers. Simultaneously, the high chromium concentration in Ni-based alloy was the reason for increasing the hardness and Young’s modulus as well as diminishing the fracture toughness [14]. The phase composition of nickel alloy also influenced the corrosion resistance of borided samples, due to the formation of multiphase layers [15].

The effects of the chemical composition of nickel-based alloys on the microstructure and some mechanical properties of boride layers, produced using the plasma paste boriding, have been described in detail [14,15]. However, there is no information in the literature data reported about the wear resistance or the growth kinetics of boride layers and activation energy required for their formation on Ni-based alloys which were subjected to plasma paste boriding. Therefore, in the present study, the growth kinetics of the boride layers, produced on Nimonic 80A-alloy using this boriding technique with various processing parameters, was studied. The temperature ranged from 1023 K to 1123 K, and process duration from 3 to 6 h. The microstructure, phase composition, microhardness, wear resistance and cohesion of the produced layers were also investigated.

## 2. Materials and Methods

### 2.1. Material and Specimens’ Preparation

The selected experimental substrate material was Nimonic 80A alloy with the following chemical composition: 19.52% Cr, 0.01% Mn, 0.01% Cu, 0.25% Fe, 2.55% Ti, 0.09% Si, 0.085% C, 1.44% Al, 76.045% Ni. The flat slice-shape samples with a diameter of 25 mm and height of 7 mm were used. Before thermo-chemical treatment, the flat surfaces of the samples were ground using SiC abrasive paper (600-grit), and then cleaned with acetone and alcohol.

### 2.2. Plasma Paste Boriding

The first preparatory step for thermo-chemical treatment was coating the flat surface of the samples with boriding paste. The boron source, used for paste preparation, was borax (Na_2_B_4_O_7_). The samples coated with the paste of a thickness of 1–1.5 mm required drying at room temperature for 24 h. The devices used for plasma paste boriding were described in previous works in details [13,14,15]. The gaseous mixture used during the processes was composed of argon and hydrogen in a ratio of 1:1. Plasma paste boriding was carried out at temperature of 1023 K, 1073 K and 1123 K for 3 h, 4 h and 6 h. The process was carried out under glow discharge conditions with a potential difference of 300–350 V using a DC (direct current) power supply and under a constant pressure of 5 mbar (500 Pa). After boriding, the samples were cooled in the vacuum chamber under a protective argon atmosphere.

### 2.3. Microstructure and Properties Characterization

Directly after plasma paste boriding, the phase analysis was carried out using X-ray diffraction (XRD). For this study, the PANalytical EMPYREAN X-ray diffractometer (Malvern Panalytical Ltd., Poznan, Poland) was used. XRD patterns were obtained using a Cu Kα radiation.

In order to prepare samples for cross-sectional analysis, plasma paste borided samples were first cut out and then mounted in a conductive resin. The metallographic samples were ground using abrasive papers, polished using aluminum oxide paste and, finally, etched with Marble’s reagent. The boride layer microstructure and thickness were investigated using a scanning electron microscope (SEM) Vega 5135 (TESCAN, Poznan, Poland). The thicknesses of plasma paste borided layers were calculated as an average value from 50 measurements carried out in different areas in the cross-section of metallographic samples.

Microhardness profiles were investigated using a Micromet II hardness tester (Buehler, Poznan, Poland) under a load of 10 gf (0.098 N) and a peak-load contact of 15 s. The Vickers diamond indenter was used for this study.

Wear resistance tests were performed under conditions of dry friction using the following parameters: load of 5 kgf (49 N), counter-specimen speed of 0.26 m/s, duration of 4 h. Two specimens were investigated: non-borided Nimonic 80A-alloy and Nimonic 80A-alloy that was plasma paste borided at 1123 K for 6 h. The frictional pair consisted of a tested specimen in the shape of a plate with dimensions 12 mm × 12 mm × 4 mm, and a ring-shaped counter-specimen with the external diameter of 20 mm. The counter-specimen was made of quenched and low-temperature tempered 100CrMnSi6-4 bearing steel of a hardness of 64 HRC. During the tests, the tested specimen was immobile, whereas the counter-specimen was rotated with a speed of 250 min^−1^. Specimen and counter-specimen weights were measured every half hour during the 4 h test. Wear resistance was calculated as a mass wear intensity factor *I_mw_* according to the following equation:(1)$$I_{mw}=\frac{∆m}{S\cdot t}$$
where Δ*m* is mass loss (mg), *S* is friction surface (cm^2^), and *t* is friction time (h).

Wear behavior of specimens and counter-specimens was also evaluated using a relative mass loss Δ*m*/*m_i_* according to the equation:(2)$$\frac{∆m}{m_{i}}=\frac{m_{i}-m_{f}}{m_{i}}$$
where Δ*m* is mass loss (mg), *m_i_* is initial mass of specimen or counter-specimen (mg), and *m_f_* is final mass of specimen or counter-specimen (mg).

After the wear tests were finished, the worn surfaces of a specimens were observed with an optical microscope (OM) LAB-40 (OPTA-TECH, Poznan, Poland).

The cohesion of the plasma paste borided layer, produced at a temperature of 1123 K for 6 h, with the substrate material was evaluated according to the VDI 3198 standard [18]. This destructive method uses the standard Rockwell C method to induce massive plastic deformation of the borided layer. The generated failures show the cohesion level of the layer. Generally, according to the VDI 3198 standard, cohesion could be classified as sufficient (quality maps HF1–HF4) or insufficient (quality maps HF5–HF6). After the cohesion tests, the indentation craters with the generated damage were observed using an optical microscope.

## 3. Results and Discussion

### 3.1. Microstructure of Boride Layers

Boride layers were successfully formed on the Nimonic 80A-alloy using the plasma paste boriding. SEM micrographs of the cross-sections of the borided samples using various processing parameters are presented in Figure 1, whereas the XRD patterns of borided samples were shown in Figure 2. The morphology of the borided layers resulted from the presence of alloying elements in the substrate material. Nimonic 80A-alloy is a chromium-rich alloy. Therefore, during boriding the atoms of chromium could be a diffusion barrier hindering the diffusion of boron atoms into the substrate material during the plasma paste boriding process [19,20,21,22]. The visible effect of the hindered diffusion of boron atoms was the characteristic smooth interface between the borided layers and the substrate material. Such a situation was independent of the processing parameters (Figure 1). Simultaneously, the significant effect of processing temperature and duration on the thickness of the borided layers was observed.

The average thickness of plasma paste borided layer increased with increasing processing temperature and duration. The layers, produced at the temperature of 1023 K, were characterized by average thickness from 19.06 μm to 29.34 µm with increasing processing time. The increase in the boriding temperature to 1073 K resulted in obtaining higher layer depths, from 40.23 µm to 56.97 µm. The use of an even higher temperature of 1123 K caused a further increase in the thickness of the borided layers. Hence, the average layer depths were 55.81 µm, 66.04 µm and 77.81 µm, for the processing time of 3 h, 4 h, and 6 h, respectively.

All the produced layers consisted of various types of nickel borides and chromium borides. The XRD patterns obtained for the plasma paste borided Nimonic 80A-alloy at different processing parameters are presented in Figure 2. The results of phase analysis are provided only for extreme values of treatment time, i.e., 3 and 6 h. In the case of the processing temperature of 1023 K, the produced layers contained chromium borides (Figure 2a,b). However, the quantity and intensity of the peaks corresponding to chromium borides (CrB or Cr_2_B) were lower compared to those visible in the diffraction patterns of the samples that were borided at higher temperatures (Figure 2c–f). Both samples borided at 1023 K contained nickel borides Ni_2_B, Ni_3_B and Ni_4_B_3_. However, in the case of the process, carried out for 6 h, the boron-rich nickel boride NiB was also produced. The increase in processing temperature still resulted in the formation of both types of chromium borides (CrB and Cr_2_B) with the increased quantity and intensity of peaks corresponding to these phases. Simultaneously, in the case of the highest boriding temperature of 1123 K, all the types of nickel borides (NiB, Ni_2_B, Ni_3_B, Ni_4_B_3_) were produced (Figure 2e,f).

### 3.2. Diffusion Model

#### 3.2.1. The Integral Diffusion Model

The integral diffusion model [23] has been used to investigate the growth kinetics of the entire boride layer thickness. It was assumed that the boride layers were composed of a mixture of nickel borides (Ni_4_B_3_, NiB, Ni_2_B and Ni_3_B) with some percentage of chromium borides (CrB and Cr_2_B). The temperature range of the diffusion process is far below from the melting point of the Nimonic 80A-alloy. So, this alloy does not undergo any phase transformation during the boriding treatment. The choice of the three temperatures (1023 K, 1073 K and 1123 K) is related to the nature of the boriding process, called the plasma paste boriding process. This process has the advantage of using lower temperatures compared to the conventional boriding processes (e.g., powder-pack boriding or gas boriding) to generate the boride layers due to activation of plasma.

In this kinetic approach, the boride incubation periods were neglected. A schematic representation of the boron concentration–depth profile inside the entire boride layer is shown in Figure 3. The term C_ads_ represent the adsorbed quantity of boron atoms at the material surface [24]. C_up_ denotes the maximum boron content in the entire boride layer (=11.615 wt.%) and C_low_ is the minimum boron content in the same layer (=6.00 wt.%) according to the Boron–Nickel binary phase diagram [25]. The variable *x(t) = u* is relative to the entire boride layer thickness. C_0_ is the boron solubility in the matrix which is extremely low.

The initial and boundary conditions for the diffusion problem are given by:(3)$$t=0,\;x>0,\;\mathrm{with}\;C\left(x,t=0\right)=C_{0}\approx 0\;\mathrm{wt}.\%$$

Boundary conditions:(4)$$C\left(x=0,\;t=0\right)=C_{up}\;\mathrm{for}\;C_{ads}>6.00\;\mathrm{wt}.\%$$
(5)$$C\left(x=u\left(t\right),\;t=t\right)=C_{low}\;\mathrm{for}\;C_{ads}<6.00\;\mathrm{wt}.\%$$

The boron concentration-depth profile inside the entire boride layer is parabolic following the Goodman’s method [26]:(6)$$C\left(x,t\right)=C_{low}+a\left(t\right)\left(u\left(t\right)-x\right)+b\left(t\right){\left(u\left(t\right)-x\right)}^{2}\;\mathrm{for}\;0\leq x\leq u$$

The three time-dependent unknowns *a*(*t*), *b*(*t*) and *u*(*t*) must fulfil the boundary conditions provided by Equations (4) and (5). The two parameters *a*(*t*) and *b*(*t*) should be positive for the applicability of the integral diffusion model.

This diffusion problem is based on the differential algebraic system given by Equations (7)–(9):(7)$$a\left(t\right)u\left(t\right)+b\left(t\right)u{\left(t\right)}^{2}=\left(C_{up}-C_{low}\right)$$
(8)$$\frac{u{\left(t\right)}^{2}}{2}\frac{da\left(t\right)}{dt}+a\left(t\right)u\left(t\right)\frac{du\left(t\right)}{dt}+\frac{u{\left(t\right)}^{3}}{3}\frac{db\left(t\right)}{dt}+b\left(t\right)u{\left(t\right)}^{2}\frac{du\left(t\right)}{dt}=2Db\left(t\right)u\left(t\right)$$
(9)$$\left(C_{up}+C_{low}\right)b\left(t\right)=a{\left(t\right)}^{2}$$

In order to obtain the numerical solution of DAE (differential algebraic equations) systems defined by the Equations (7)–(9), there was a need to establish the initial conditions which should be consistent. So, the initial conditions were given above. For the numerical resolution of the obtained DAE system, the freely available software called the Interactive Thermodynamics software (version 3.2) was used.

To obtain the expression of boron diffusion coefficient through the entire boride layer, the following change of variables was considered:(10)$$u\left(t\right)=2\varepsilon \sqrt{Dt}$$
(11)$$a\left(t\right)=\frac{a}{u\left(t\right)}$$
and
(12)$$b\left(t\right)=\frac{\beta }{u{\left(t\right)}^{2}}$$
where *u*(*t*) is the entire boride layer thickness, and *ε* is non-dimensional parameter to be determined.

After some mathematical manipulations (substitution and derivation) in the obtained DAE system [27], it is possible to estimate the three unknowns *α*, *β* and *ε*.
(13)$$\alpha =\frac{\left(C_{up}+C_{low}\right)}{2}\left[-1+\sqrt{1+4\left(\frac{C_{up}-C_{low}}{C_{up}+C_{low}}\right)}\right]$$
(14)$$\beta =\frac{\alpha^{2}}{\left(C_{up}+C_{low}\right)}$$
and
(15)$$\varepsilon =\sqrt{\frac{6\beta }{3\alpha +2\beta }}=\sqrt{\frac{3\left[-1+\sqrt{1+4\frac{\left(C_{up}-C_{low}\right)}{\left(C_{up}+C_{low}\right)}}\right]}{2+\sqrt{1+4\frac{\left(C_{up}-C_{low}\right)}{\left(C_{up}+C_{low}\right)}}}}$$

The term $k=2\varepsilon \sqrt{D}$ represents the slope of the experimental curve giving the variation of entire boride layer thickness as a function of square root of time. So, it is easy to deduce the value of boron diffusion coefficient *D* through the entire boride layer as follows:(16)$$D={\left(\frac{k}{2\varepsilon }\right)}^{2}$$
with *ε* = 0.6593.

#### 3.2.2. Calculation of Boron Diffusion Coefficients through the Entire Boride Layer

The experimental data in terms of kinetics were analyzed by plotting the evolution of entire boride layer thickness versus the square root of treatment time at 1023, 1073 and 1123 K as shown in Figure 4. Table 1 lists the experimental values of parabolic growth constants in the temperature range of 1023–1123 K.

The boron diffusion coefficient *D* inside the entire boride layer can be obtained from Equation (16) by using the data given in Table 1. Table 2 gives the calculated values of boron diffusion coefficients by the integral diffusion model with a maximum boron concentration of 11.615 wt.%.

The value of boron activation energy can be readily obtained from the slope of the curve relating the natural logarithm of boron diffusivity versus the reciprocal temperature (Figure 5). As a result, the expression of boron diffusion coefficient *D* is given by Equation (17) with a coefficient of determination of 0.976:(17)$$D=1.392\cdot 10^{-4}exp\left(\frac{-190.93\frac{\mathrm{kJ}}{\mathrm{mol}}}{RT}\right)m^{2}/s$$
when *R*^2^ = 0.976, *T* is the absolute temperature (K), and *R* is the ideal gas constant (*R* = 8.314 J·mol^−1^·K^−1^).

#### 3.2.3. Comparison of Boron Activation Energy with the Literature Results

The values of boron activation energies in nickel 201 alloy and Ni_3_Al substrates [28,29,30,31] are listed in Table 3, together with the values obtained on Nimonic 80 A alloy. It can be observed that the reported values of boron activation energies are affected by the following factors: the boriding technique, the chemical composition of material substrate, the method of computation.

For instance, Calik et al. [28] studied the powder-pack boriding of the pure nickel in the temperature range 1123–1273 K. They obtained a value of 47.3 kJ·mol^−1^, which is very low compared with the data given in Table 3. This result is incompatible with other literature data [29,30,31]. Furthermore, Kahvecioglu et al. [30] used an ultra-fast electrochemical boriding for nickel aluminide between 1073 and 1223 K by changing the value of current density from 0.1 to 0.5 A·cm^−2^. They obtained a value of boron activation energy in Ni_3_Al equal to 185.96 kJ·mol^−1^. Torun [31] investigated the powder-pack boriding of the Ni_3_Al substrate between 1073 and 1223 K. The estimated value of boron activation energy in this substrate was found to be equal to 188 ± 14.4 kJ·mol^−1^. The obtained value of boron activation energy in Nimonic 80A-alloy in the present study was in line with the data [28,29,30,31,32] listed in Table 3.

In all the above-mentioned papers [28,29,30,31], including the present work, the activation energy was calculated taking into consideration the growth kinetics of the total boride layers, containing the mixture of borides. In one paper [32], the zonation of the boride layers, produced on Inconel 718-alloy by powder-pack boriding, was assumed. The presence of the three successive boride zones (Ni_4_B_3_, Ni_2_B and Ni_3_B) with the increase in distance from the surface was considered in the cross-section of borided layers. The heat balance integral model was proposed to describe the growth kinetics of such layers. As a consequence, the boron activation energy depended on the zone considered, obtaining the slightly higher values when compared to the papers [29,30,31] or the present study (see Table 3).

#### 3.2.4. Prediction of Entire Boride Layer Thickness Using the Numerical Solution of DAE System

The experimental and predicted values of entire boride layer thickness for a maximum boron concentration equal to 11.615 wt.% are gathered in Table 4.

The estimation of layers’ thicknesses by the integral diffusion model was carried out via a numerical solution of DAE system [27] by taking as initial conditions *a_0_* = 44.77, *b_0_* = 113.791 and *u_0_* = 0.10 µm with the help of a computer simulation program (using the Interactive Thermodynamics software version 3.2) It is seen that the predicted values of layers’ thicknesses are in line with the predicted results (see Table 4).

### 3.3. Microhardness Profiles

The microhardness measurements were carried out in the cross-section of the plasma paste borided layers up to the substrate material. The microhardness profiles presented the Vickers microhardness values as a function of the distance from the surface for the Nimonic 80A-alloy which was plasma paste borided with different processing parameters. The results are shown only for extreme values of treatment duration, i.e., 3 and 6 h (Figure 6).

Due to the multiphase character of the produced layers, which contained a mixture of various nickel and chromium borides, the microhardness measured in the borided layers ranged from 1160 HV to 2132 HV. Such a situation was expected, because it was a characteristic dependence of hardness on the phase composition of the tested area, often occurring in the case of a multiphase structure. In general, the presence of chromium borides in the microstructure of borided layers, produced on nickel alloys, caused an increase in hardness. Such a behavior was observed in the case of nickel alloys borided using gaseous technique [33,34] and laser alloying with boron [35,36]. In the case of plasma paste boriding, the applied processing parameters slightly influenced the phase composition of the surface layers formed on Nimonic 80A-alloy. All the produced layers were composed of a mixture of nickel and chromium borides. However, some differences were visible in the intensity of the peaks corresponding to the chromium borides (CrB and Cr_2_B). In the case of the lowest boriding temperature (1023 K), the quantity and intensity of these peaks were lower (Figure 2a,b) compared to those presented in the diffraction patterns obtained for higher boriding temperatures (Figure 2c–f). Probably, these differences caused a slight decrease in the maximum hardness of the boride layers produced at 1023 K for 3 and 6 h. It is clearly visible in Figure 6a,b, respectively.

### 3.4. Cohesion

The standard Rockwell C hardness test was applied for cohesion evaluation. The Rockwell C indentation craters with failures produced on the surface of Nimonic 80A-alloy, plasma paste borided at 1123 K for 6 h, are shown in Figure 7. In general, the cohesion of the produced boride layer into the substrate material can be classified as a sufficient. Based on the cohesion quality maps in the VDI 3198 standard [18], the more detailed classification of the cohesion level can be determined. The Rockwell indentation under a load of 150 kgf (1471 N) caused the production of a multiple cracking system in a form of radial cracks (Figure 7a,b). Simultaneously, the presence of areas with flaking and delamination was also detected using higher magnifications of the optical microscope (Figure 7c,d). The type and quantity of the identified failures indicated the cohesion corresponding to the HF3 standard.

### 3.5. Wear Resistance

The tribological properties of the plasma paste borided Nimonic 80A-alloy were compared to those of the non-borided Nimonic 80A-alloy. The plasma paste borided layer, produced on Nimonic 80A-alloy at 1123 K for 6 h, was investigated. The wear resistance was determined in two ways: by calculating the mass wear intensity factor *I_mw_* (Figure 8a) and by calculating the relative mass loss (Figure 8b). The lower value of *I_mw_*, the better wear resistance of the studied material.

The analysis of mass wear intensity factors indicated the improved wear resistance of the plasma paste borided sample (*I_mw_* = 1.47 mg/cm^2^) in comparison to the non-borided sample (*I_mw_* = 9.06 mg/cm^2^). A significant reduction in the relative mass loss Δ*m/m_i_* of the plasma paste borided Nimonic 80A-alloy was demonstrated (Figure 8b). The calculated Δ*m/m_i_* ratio for the non-borided Nimonic 80A-alloy was 2.5-times higher compared to the plasma paste borided Nimonic 80A-alloy. The produced borided layer contained a mixture of nickel and chromium borides of high hardness. Therefore, an intense mass loss of the counter-specimen (quenched and low-temperature tempered 100CrMnSi6-4 bearing steel) was visible for the friction pair consisting of the borided Nimonic 80A-alloy and counter-specimen. The worn surfaces of borided and non-borided samples after the wear tests were observed using an optical microscope and shown in Figure 8c,d, respectively. The worn surface of the plasma paste borided Nimonic 80A-alloy was characterized by the presence of numerous shallow grooves. Simultaneously, no signs of plastic deformation and adhesion were visible (Figure 8c). On the contrary, the surface of the non-borided Nimonic 80A-alloy was characterized by obvious signs of severe plastic deformation after the wear resistance test (Figure 8d).

## 4. Summary and Conclusions

The plasma paste boriding process was used in order to produce the multi-phase boride layers on Nimonic 80A-alloy. The process was carried out in a H_2_-Ar gas mixture using borax paste as a boron source. The treatment temperature range of 1023–1123 K and treatment time of 3 h, 4 h and 6 h were applied during plasma paste boriding processes. Growth kinetics and some mechanical properties of plasma paste borided Nimonic 80A-alloy were studied using the integral diffusion model. Based on the detailed analysis of the results, the following conclusions could be formulated:The increased processing temperature and longer processing duration ensured the production of thicker borided layers;All the produced layers consisted of a mixture of nickel borides and chromium borides;The quantity and intensity of the peaks corresponding to chromium borides (CrB or Cr_2_B) increased at elevated boriding temperature;The theoretical thicknesses of the borided layers were estimated using an integral diffusion model. A good correlation was obtained between the theoretical (modeled) and experimental depths of the plasma paste borided layers;The estimated value of boron activation energy was found to be equal to 188 ± 14.4 kJ·mol^−1^. This value was in line with the data of other papers which considered the growth kinetics of the total boride layers, containing the mixture of borides;The produced layers were characterized by high hardness ranging from 1160 HV to 2132 HV. It resulted from the multiphase character of the produced layers, which contained a mixture of various nickel borides and chromium borides of different hardness;The cohesion of produced boride layers into the substrate material can be classified as a sufficient, corresponding to the HF3 quality map;The analysis of mass wear intensity factors indicated the improved wear resistance of the plasma paste borided sample (*I_mw_* = 1.47 mg/cm^2^) in comparison to the non-borided sample (*I_mw_* = 9.06 mg/cm^2^). A significant reduction in the relative mass loss Δ*m/m_i_* of the plasma paste borided Nimonic 80A-alloy was also demonstrated.

## Figures and Tables

**Figure 1 materials-14-05146-f001:**
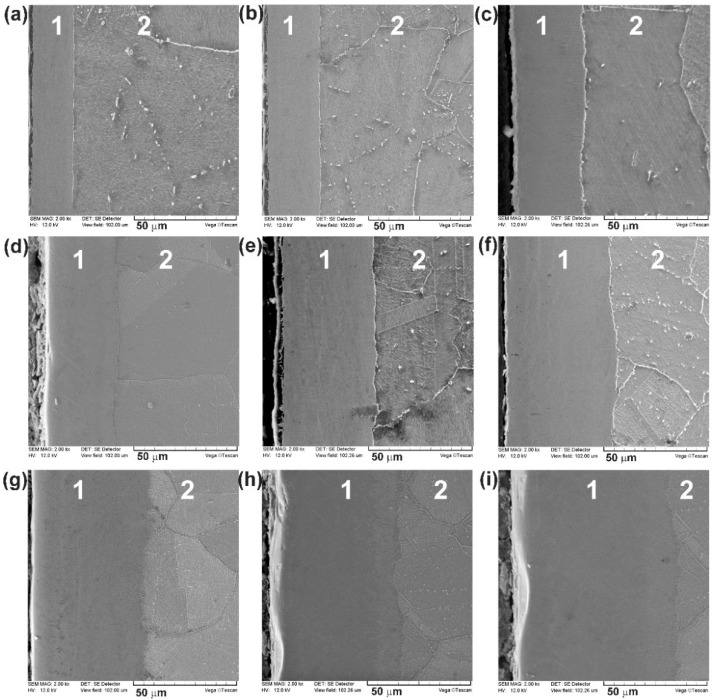
Microstructure of Nimonic 80A-alloy after plasma paste boriding using various parameters: (**a**) 1023 K for 3 h, (**b**) 1023 K for 4 h, (**c**) 1023 K for 6 h, (**d**) 1073 K for 3 h, (**e**) 1073 K for 4 h, (**f**) 1073 K for 6 h, (**g**) 1123 K for 3 h, (**h**) 1123 K for 4 h, (**i**) 1123 K for 6 h; 1—compact boride layer, 2—substrate material.

**Figure 2 materials-14-05146-f002:**
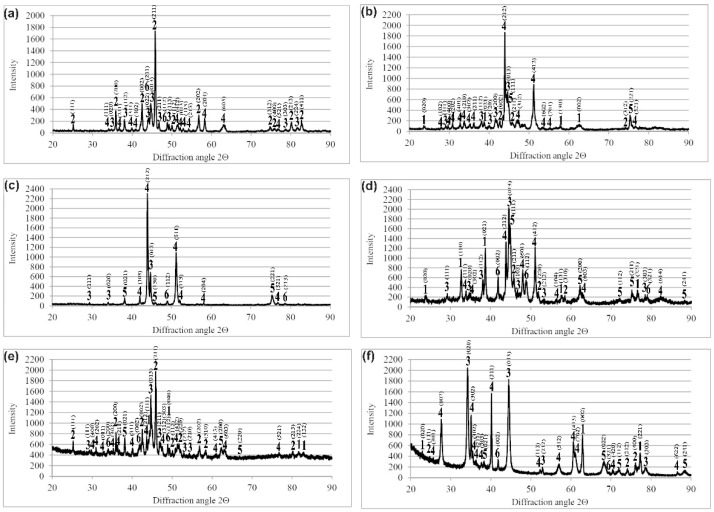
XRD patterns of Nimonic 80A-alloy after plasma paste boriding using selected parameters: (**a**) 1023 K for 3 h, (**b**) 1023 K for 6 h, (**c**) 1073 K for 3 h, (**d**) 1073 K for 6 h, (**e**) 1123 K for 3 h, (**f**) 1123 K for 6 h; 1—NiB, 2—Ni_2_B, 3—Ni_3_B, 4—Ni_4_B_3_, 5—CrB, 6—Cr_2_B.

**Figure 3 materials-14-05146-f003:**
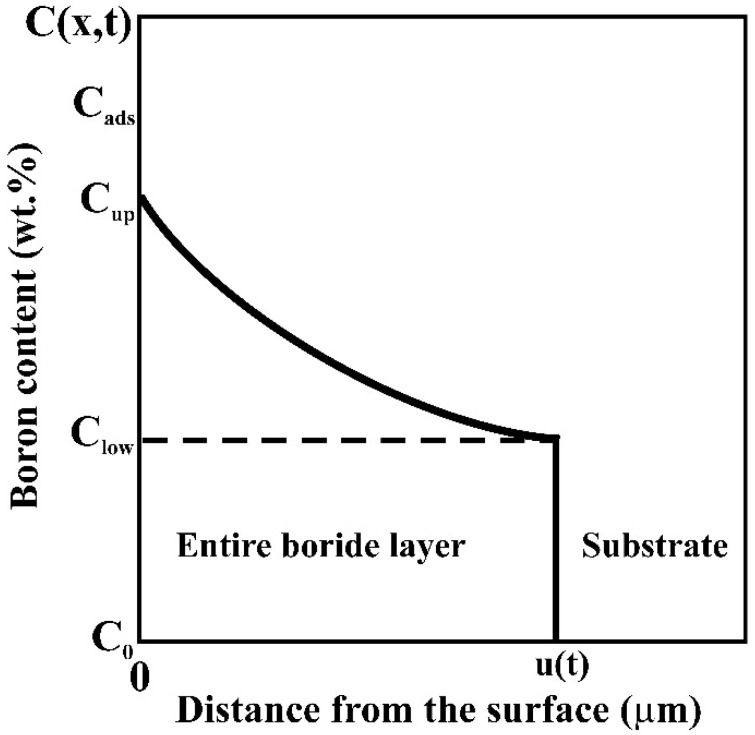
Schematic representation of boron concentration-depth profile inside the entire boride layer in a saturated substrate with boron atoms.

**Figure 4 materials-14-05146-f004:**
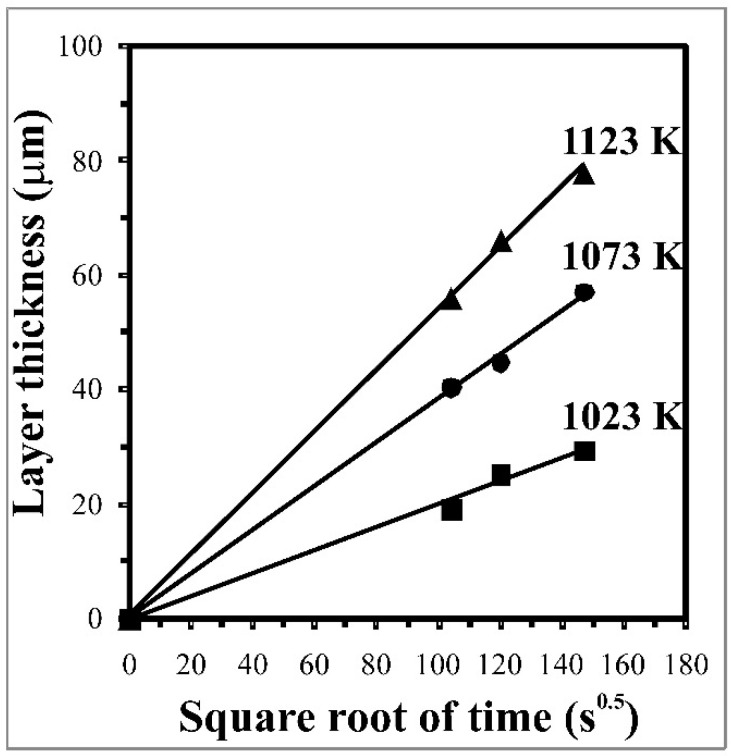
Time dependence of entire boride layer thickness for increasing temperatures.

**Figure 5 materials-14-05146-f005:**
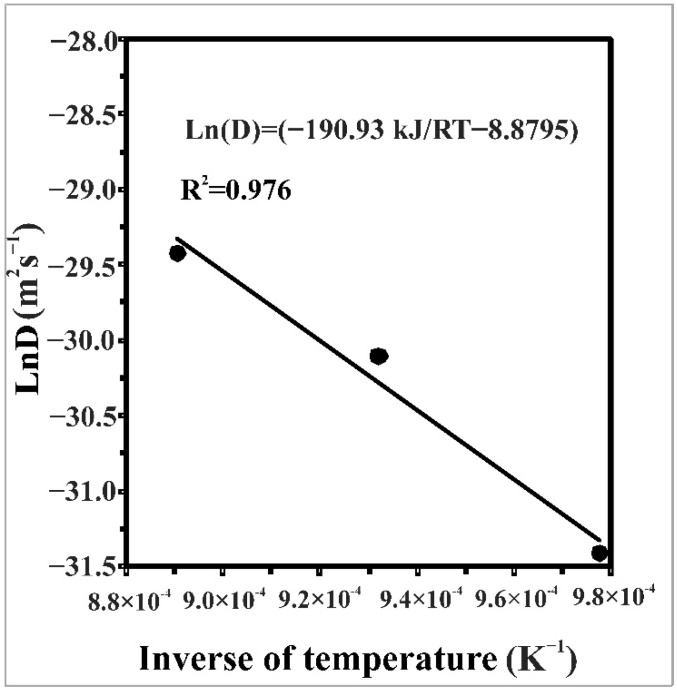
Temperature dependence of *D* versus the inverse of temperature.

**Figure 6 materials-14-05146-f006:**
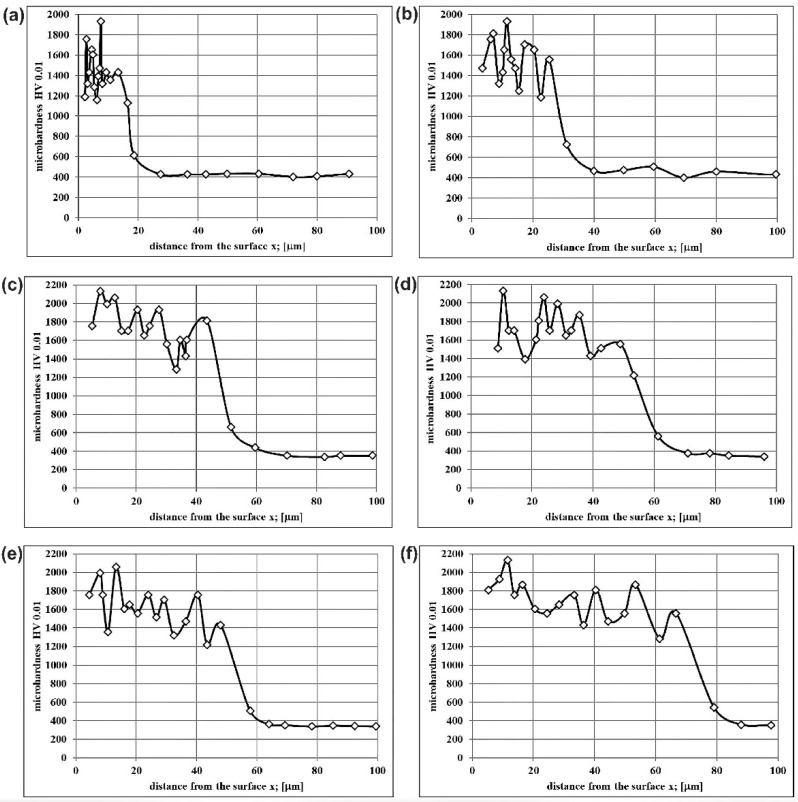
Microhardness profiles of Nimonic 80A-alloy after plasma paste boriding using selected parameters: (**a**) 1023 K for 3 h, (**b**) 1023 K for 6 h, (**c**) 1073 K for 3 h, (**d**) 1073 K for 6 h, (**e**) 1123 K for 3 h, (**f**) 1123 K for 6 h.

**Figure 7 materials-14-05146-f007:**
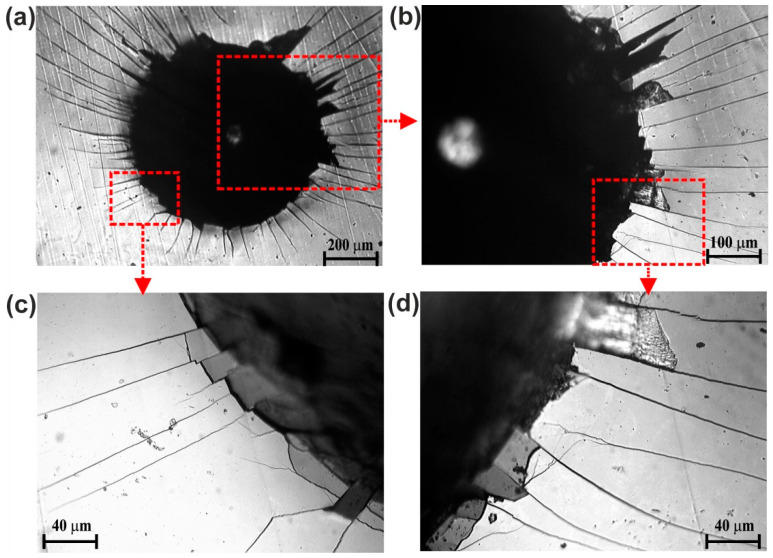
Rockwell C indentation craters with failure marks produced on the surface of plasma paste borided Nimonic 80A-alloy at 1123 K for 6 h; the entire crater (**a**), enlargement of selected fragments of the crater (**b**–**d**).

**Figure 8 materials-14-05146-f008:**
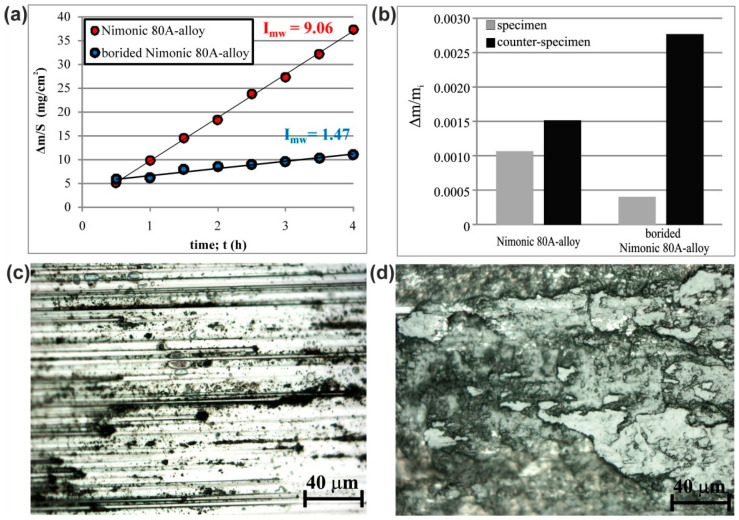
Results of wear resistance tests: (**a**) mass loss on a unit of friction surface vs. time of friction with calculated *I_mw_* factors, (**b**) relative mass loss of specimens and counter-specimens, (**c**) worn surface of plasma paste borided Nimonic 80A-alloy, (**d**) worn surface of non-borided Nimonic 80A-alloy.

**Table 1 materials-14-05146-t001:** Experimental values of parabolic growth constants in the temperature range of 1023–1123 K.

Temperature (K)	*k* (µm s^−0.5^) at the Entire Boride Layer/Substrate Interface
1023	0.1987
1073	0.3825
1123	0.5376

**Table 2 materials-14-05146-t002:** Estimated values of boron diffusion coefficients by the integral diffusion model with an upper boron content of 11.615 wt.%.

Temperature (K)	*D* (×10^−14^ m^2^/s)
1023	2.27
1073	8.41
1123	16.62

**Table 3 materials-14-05146-t003:** Boron activation energies in pure nickel and its alloys depending on the boriding technique.

Material	Boriding Technique	Temperature Range (K)	Phases Present in the Boride Layer	Boron Activation Energy (Kj·Mol^−1^)	Calculation Method	Refs.
Pure Ni	Powder-pack boriding (Ekabor II)	1123–1273	Ni_2_B, Ni_6_Si_2_B	47.3	Parabolic growth law	[28]
Nickel 201	Powder-pack boriding (B_4_C + KBF_4_)	1123–1273	NiB, Ni_2_B, Ni_3_B, Ni_4_B_3_	203.87	Parabolic growth law	[29]
Ni_3_Al	Electrochemical boriding	1073–1223	Ni_3_B, Ni_4_B_3_, Ni_20_AlB_14_	185.95	Parabolic growth law	[30]
Ni_3_Al	Powder-pack boriding (Ekabor-Ni)	1073–1223	Ni_3_B, Ni_4_B_3_, Ni_3_Al	188 ± 14.4	Parabolic growth law	[31]
Inconel 718	Powder-pack boriding (90% B_4_C + 10% KBF_4_)	1123–1223	Ni_4_B_3_, Ni_2_B, Ni_3_B, Fe_2_B, Cr_2_B	233.20 (Ni_4_B_3_) 206.17 (Ni_2_B) 218.06 (Ni_3_B)	Integral diffusion model	[32]
Nimonic 80A	Plasma paste boriding	1023–1123	NiB, Ni_2_B, Ni_3_B, Ni_4_B_3_	190.93	Integral diffusion model	[The present work]

**Table 4 materials-14-05146-t004:** Experimental and simulated values of entire boride layers’ thicknesses.

Temperature (K)	Time (h)	Experimental Layer Thickness (µm)	Standard Deviation (mm)	Predicted Layer Thickness (µm)
1023	0	0	0	0
	3	19.06	0.67	21.58
	4	25.06	0.61	24.92
	6	29.34	1.73	30.52
1073	0	0	0	0
	3	40.23	0.82	36.41
	4	44.56	1.28	42.04
	6	56.97	2.3	51.49
1123	0	0	0	0
	3	55.81	0.85	58.63
	4	66.04	1.28	67.7
	6	77.82	3.09	82.93

## Data Availability

The authors confirm that the data supporting the findings of this study is available within the article.

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
