# Peer review of "Growth Kinetics and Some Mechanical Properties of Plasma Paste Borided Layers Produced on Nimonic 80A-Alloy"

_materials, 2021, doi:10.3390/ma14185146_

Round 1

Reviewer 1 Report

The manuscript reports on the growth kinetics and some mechanical properties of plasma paste borided layers produced on Nimonic 80A-alloy. The presented results would be interesting to materials communities. Thus, the manuscript can be recommended to be published in Materials after the authors addressing the following comments/concerns.

Introduction

  • Perhaps it is worth adding information on why nickel alloys were taken for research? Why was this particular alloy chosen?

Materials and Methods

  • Lines 118 and 120 - In the text, the information that must be indicated in the form of superscripts / subscripts goes in the main text.

Results and Discussions

  • Line 158 - Perhaps it is worth increasing the captions on the scale bar (Fig. 1)? What is the error in determining the layer thickness?
  • Line 186 - It is difficult to distinguish the signatures of the reflections in Figure 2. It is worth pointing out the difference between different samples obtained at the same temperature, including the description of the samples at 4 h.
  • Lines 200-203 – repetition of 1 paragraph of section 3.2.1 on page 5.
  • Line 197 - Figure 3 should be placed after the mention in the text.
  • Lines 207 and 208 - the information that must be indicated in the form of superscripts / subscripts goes in the main text (eg, Cup, Clow).
  • Line 264 - Figure 5 should be placed after the mention in the text.
  • Missing reference 27 in the text?
  • Line 309 - Does this reference (27) appear in the text after [31]?

However, these comments do not reduce the relevance and importance of the results. The article is sufficiently novel and interesting to warrant publication. Results and discussion are reliable. References reflect the main publications on which the work is based.

Reviewer 2 Report

The authors have carried out significant experimental work to study the growth kinetics and mechanical properties of plasma paste borated layers. The manuscript can be recommended for publication after making minor corrections.

  1. What is the reason for the choice of the temperatures 1023K, 1073K and 1123K? How do these temperatures relate to the melting temperature of the substrate?
  2. The time t must be present in the denominator of the right part of equation (16).
  3. Authors should understand that Eq. (17) is not accurate, since it is difficult to construct a reliable relationship between physical quantities based only three points in LnD vs. 1/T plot (see Fig. 5).
  4. Page 10, section 3.2.4: Authors should describe in more detail the computer simulation procedure for the prediction of the entire boride layer thickness.
